# PLEKHM2 Loss of Function Impairs the Activity of iPSC-Derived Neurons via Regulation of Autophagic Flux

**DOI:** 10.3390/ijms232416092

**Published:** 2022-12-17

**Authors:** Hadas Ben-Zvi, Tatiana Rabinski, Rivka Ofir, Smadar Cohen, Gad D. Vatine

**Affiliations:** 1The Avram and Stella Goldstein-Goren Department of Biotechnology Engineering, Faculty of Engineering Sciences, Ben-Gurion University of the Negev, Beer Sheva 84105, Israel; 2The Regenerative Medicine and Stem Cell (RMSC) Research Center, Ben-Gurion University of the Negev, Beer Sheva 84105, Israel; 3Dead Sea & Arava Science Center, Masada 8691000, Israel; 4Ilse Katz Institute for Nanoscale Science and Technology, Ben-Gurion University of the Negev, Beer Sheva 84105, Israel; 5The Department of Physiology and Cell Biology, Faculty of Health Sciences, Ben-Gurion University of the Negev, Beer Sheva 84105, Israel; 6The Zelman School of Brain Sciences and Cognition, Ben-Gurion University of the Negev, Beer Sheva 84105, Israel

**Keywords:** neurons, motor neurons, autophagy, PLEKHM2, DCM-LVNC, neurodegeneration, lysosomes, autophagosomes, iPSCs, disease model

## Abstract

Pleckstrin Homology And RUN Domain Containing M2 (PLEKHM2) [delAG] mutation causes dilated cardiomyopathy with left ventricular non-compaction (DCM-LVNC), resulting in a premature death of PLEKHM2[delAG] individuals due to heart failure. PLEKHM2 is a factor involved in autophagy, a master regulator of cellular homeostasis, decomposing pathogens, proteins and other cellular components. Autophagy is mainly carried out by the lysosome, containing degradation enzymes, and by the autophagosome, which engulfs substances marked for decomposition. PLEKHM2 promotes lysosomal movement toward the cell periphery. Autophagic dysregulation is associated with neurodegenerative diseases’ pathogenesis. Thus, modulation of autophagy holds considerable potential as a therapeutic target for such disorders. We hypothesized that PLEKHM2 is involved in neuronal development and function, and that mutated PLEKHM2 (PLEKHM2[delAG]) neurons will present impaired functions. Here, we studied PLEKHM2-related abnormalities in induced pluripotent stem cell (iPSC)-derived motor neurons (iMNs) as a neuronal model. PLEKHM2[delAG] iMN cultures had healthy control-like differentiation potential but exhibited reduced autophagic activity. Electrophysiological measurements revealed that PLEKHM2[delAG] iMN cultures displayed delayed functional maturation and more frequent and unsynchronized activity. This was associated with increased size and a more perinuclear lysosome cellular distribution. Thus, our results suggest that PLEKHM2 is involved in the functional development of neurons through the regulation of autophagic flux.

## 1. Introduction

Autophagy is one of the systems entrusted to maintain homeostasis in cells and organs [1,2]. Macro-autophagy (hereafter autophagy) is carried out by two principal organelles, the autophagosome and the lysosome, along with an abundance of supporting protein complexes. The autophagosome engulfs proteins, pathogens and other cellular components, which are labeled for degradation. The lysosome is mounted with hydrolases that are active in acidic pH and their fusion, namely autolysosomes, results in the decomposition of the autophagosome content and recycling of the individual amino acids obtained [3,4]. Recently, a novel mutation in the *Pleckstrin Homology And RUN Domain Containing M2* (*PLEKHM2*) gene, *PLEKHM2*[delAG], was linked to a rare and fatal cardiovascular disease called dilated cardiomyopathy with left ventricular non-compaction (DCM-LVNC). This horrid disease was shown to progress due to autophagy impairments [5]. PLEKHM2 (also known as SKIP) assists with anterograde lysosomal movement, which transports organelles from the nucleus to the cell periphery by traveling across the microtubules utilizing KINESIN-1 motor protein [6,7]. Many adaptors, including PLEKHM2, are responsible for proper lysosomal-motor protein association ensuring long-range organelle transport is carried out [8]. In anterograde motion, KINESIN-1 binds to BLOC-1-related complex (BORC), which recruits small GTPase ARL8 (abbreviated ARL8), coupling indirectly to KINESIN-1 subunits through PLEKHM2 [6,9]. Proper lysosome function is highly important in cellular processes, such as antigen presentation [10], liver function [11] and neurotransmitter turnover [12,13]. Therefore, it is not surprising that autophagic dysregulation has been proven to accompany pathogenesis [14,15,16,17], including neurodegenerative disorders such as Parkinson’s disease (PD) [18], Alzheimer’s disease (AD) [19], and amyotrophic lateral sclerosis (ALS) [20]. Deviations in lysosomal motility and complex formation are also crucial. For example, alterations in BORC and KINESIN-1 were associated with psychiatric and neurological disorders [21]. Additionally, KINESIN-1 and ARL8 inhibition due to cholesterol accumulation on lysosomal membranes was discovered in the progressive disorder known as Niemann–Pick’s disease type C (NPC) [22]. Recently, a regulatory mechanism in which PLEKHM2 auto-inhibits its interaction with KINESIN-1 by coupling its N- and C- terminals was discovered [8]. PLEKHM2 is re-activated by ARL8, recruiting PLEKHM2 for lysosomal anterograde motion [6,8].

PLEKHM2[delAG] patients experienced premature death due to improper autophagic process. Since, mRNA levels of PLEKHM2 in human tissues had the highest expression in the brains and testis [5], and as neurons are greatly dependent on homeostasis to function properly, we hypothesized that PLEKHM2[delAG] mutation may result in neuropathogenesis. We therefore sought to study patient-specific cells as a research tool to study the involvement of PLEKHM2 on neuronal functions.

Here, we describe an in vitro model that enables us to address the role of PLEKHM2 in the context of functional neuronal maturation. Induced pluripotent stem cell (iPSCs) from two DCM-LVNC patients and from a heterozygous healthy sibling [23] were differentiated to motor neurons (iMNs), as a neuronal source. PLEKHM2[delAG] neurons in iMN cultures displayed increased lysosome size that remained in proximity to the nucleus, which was accompanied by reduced autophagic flux. In addition, the development of spontaneous electrophysiology activity was delayed and exhibited increased firing frequency and asynchrony within the neuronal network. Our findings emphasize the importance of PLEKHM2 in the development and regulation of functional neuronal maturation.

## 2. Results

### 2.1. PLEKHM2[delAG] Patients and iPSCs

Fibroblasts from two DCM-LVNC sibling patients with a homozygous (chr1:16055171_16055172delAG) disease-causing mutation and their healthy heterozygous sibling were collected and reprogrammed into patient-specific iPSCs using non-integrating Sendai virus as previously described in detail [23], officially registered as BGUi009, BGUi010 and BGUi008 in hPSCreg, respectively. The PLEKHM2 protein expressed in fibroblasts was previously suggested to be shorter in PLEKHM2[delAG] cultures [5], translated either with or without the pleckstrin homology (PH) domain (Figure 1). To assess PLEKHM2 expression in our cells, we performed a Western blot analysis using a previously validated anti-PLEKHM2 antibody targeting the N-terminal domain [5]. iPSC-derived neuroepithelial cells from both PLEKHM2[delAG] patients and controls displayed several bands including the expected 113 kDa (Appendix A). Quantifying this band showed a 2-fold decrease in protein levels, suggesting that this variant affects its protein levels. Since no additional bands were detected in the PLEKHM2[delAG] cells, we conclude that the PH-domain aberrant variant is either nonexistent or expressed at low levels in PLEKHM2[delAG] cells.

Although these patients died as young adults before they exhibited any neurological pathology phenotype, previous studies suggested that autophagy impairments correlate with neuropathology in other disorders [24,25]. Furthermore, iPSC-derived neurons were previously shown to be useful in uncovering subtle differences in both developmental and late-onset neurological disorders [26,27]. We therefore sought to take advantage of these iPSCs to generate an in vitro model for studying the role of PLEKHM2 in autophagy and to test its role in functional neuronal development.

### 2.2. PLEKHM2[delAG] iPSCs Can Differentiate into iMNs

iPSCs were differentiated into motor neurons as previously described [27] (Figure 2A). Differentiated cells highly expressed neural markers including SOX1 and the neural progenitor marker NESTIN. As expected, differentiated cells displayed reduced expression of the pluripotency marker OCT3/4 compared to undifferentiated iPSCs. Motor neuron progenitor cells (MNPCs) continued to proliferate while pluripotency decreased. Proliferation halts when applying S3M, which induces maturation, pushing MNPCs towards mature MN fate (Figure 2B–D).

Immunocytochemistry (ICC) analysis showed that neuronal markers TUJ1α, MAP2ab and ISL1 are expressed in MNPCs and more pronounced in iMNs (Figure 2E,F). Contrarily, the levels of the neural progenitor marker, SOX1 decreased during the transition to iMNs. Flow cytometry analysis confirmed that iMN cultures contain MAP2ab- and TUJ1α-positive neurons, as well as ISL1-positive motor neurons (Figure 2G). No significant differences were observed between control and PLEKHM2[delAG] cultures at all stages of differentiation (D0, D6, D12 and D18). These results suggest that PLEKHM2 is not involved in the differentiation of iPSCs to motor neurons.

### 2.3. PLEKHM2[delAG] Cultures Show Reduced Autophagic Activity

PLEKHM2 was previously shown to be required for the autophagic process [5,6,21]. Autophagic activity occurs constitutively at a low rate, but when stress is applied autophagy levels increase to restore homeostasis [28]. Therefore, we next tested autophagy in PLEKHM2[delAG] and control lines throughout the different stages of differentiation to determine whether this activity was compromised under stress. Cells were treated by either inhibiting the target of rapamycin (mTOR) pathway with rapamycin^®^ (Enzo Life Sciences, Tokyo, Japan) at a concentration of 500 nM for 20 h, or by nutrient deprivation (termed here NLM) that prompts recycling of proteins within the cell, for 3 h [29]. Chloroquine (CQ) inhibits lysosomal function as it alkalinizes lysosomal pH and as such, a combination of either R + CQ or NLM + CQ treatment is expected to exhibit an accumulation of cellular autophagosomes and lysosomes. The CYTO-ID autophagy detection kit (Enzo Life Sciences) was used to label the autophagosomes and the autophagic flux was evaluated. Flow cytometry examination of the cultures showed insignificant differences in autophagic flux throughout differentiation between control and PLEKHM2[delAG] (Figure 3A). iPSC, MNPC and iMN cultures only exhibited a modest response with changes of less than 2-fold to the different stressors in both control and PLEKHM2[delAG] cells (marked by a dashed line). On the other hand, at the NEP stage, stress induced more than 2-fold production of autophagosomes. However, no significant differences were observed between the treatments at the different stages. Since NEP cultures had the greatest response to the treatments, we next tested the protein expression of the established autophagosome markers LC3B and p62 [30,31] by Western blot. The LC3B band did not clearly separate into its two isoforms (LC3BI and LC3BII). LC3BII is the most common marker for the autophagic process [32]. We therefore calculated its levels based on the whole protein band. Interestingly, a lower production of autophagosomes provoked by both stress treatments was observed in PLEKHM2[delAG] cells. These results support the non-significant trend observed in the flow cytometry analysis (Figure 3A,B). Collectively, the data showed reduced autophagy activity in PLEKHM2[delAG] cultures when compared to the control cultures.

### 2.4. PLEKHM2[delAG] Culture Activity Peaks Later Than Its Control

To test whether PLEKHM2 is involved in functional maturation, neural cells from both PLEKHM2[delAG] and healthy control were seeded and differentiated on an MEA plate and spontaneous activity was monitored twice weekly during the course of five weeks. The spontaneous activity peaked in the control cells during the third week (D15-D21) of differentiation with an average of 15.3 spikes/min. The activity decreased and by the fifth week only 0.72 spikes/min were recorded (Figure 4A). Interestingly, in PLEKHM2[delAG] cells the neuronal activity peaked at a later stage, during the fifth week of differentiation (D30–D33), and displayed elevated activity with an average of 40 spikes/min (Figure 4A). Additionally, PLEKHM2[delAG] lines fired action potentials at an irregular rate with an increased frequency, while control lines rarely fired, but when doing so exhibited great network connectivity indicated by simultaneous firing throughout all electrodes (Figure 4B,C). Additionally, PLEKHM2[delAG] cultures showed an elevated inter-spike interval (ISI) and higher burst percentage average when compared to control cultures (Figure 4D,E). The synchrony index (ranging from 0—no synchrony to 1—highly synchronized) serves as a measure for the connectivity of neurons in a culture [33,34]. Comparing the synchrony index of weeks 3 and 5 revealed a significant difference in control and PLEKHM2[delAG] connectivity. While control cultures become more synchronized from week 3 to 5, PLEKHM2[delAG] lose their synchrony during that time period (Figure 4F). The results showed that although both patient and control lines differentiated into active neuronal cultures, PLEKHM2[delAG] cells present impaired functionality when compared to control cells.

To confirm that changes in electrophysiological activity were not due to cell detachment or cell death, on the fifth week of differentiation, cells were lifted from the MEA plates and subjected to flow cytometry analysis. Both control and PLEKHM2[delAG] cultures contained mostly neurons with about 10% astrocytes (Figure 4G). The culture also contained 10% of ChAT-positive motor neurons. No significant differences were found in the number of cells per well (Appendix A) nor in the differentiation potential between control and PLEKHM2[delAG] cells. These results suggest that the differences in the neuronal activity are not the consequence of differential cell differentiation or survival.

### 2.5. The Mutation in PLEKHM2 Leads to Lysosomal Dysfunction

PLEKHM2 participates in the association of the motor protein KINESIN-1 to the lysosome. In turn, KINESIN-1 propels anterograde movement towards the axon terminal [6,7,21]. To evaluate whether anterograde movement is PLEKHM2-dependent, lysosomes in iMN cultures were labeled by lysosomal-associated membrane protein 1 (LAMP1), a lysosomal membrane marker [35], under basal conditions and following stress induction by R + CQ. While LAMP1 marks lysosomes, it is also apparent in other cellular vesicles, mostly in late endosomes, due to their transport to lysosomes in the endocytotic pathway [9]. We therefore refer to all LAMP1-positive vesicles as lysosomes. Stress induction treatment led to a greater number of lysosomes per cell in control neurons, compared to PLEKHM2[delAG] neurons (Figure 5A). Interestingly, the lysosome size in PLEKHM2[delAG] cells is increased compared to controls by approximately 38% under basal conditions and by 21% following treatment (Figure 5B). Additionally, the cellular distribution of lysosomes was affected by the mutation (Figure 5C); PLEKHM2[delAG] neurons showed a perinuclear localization of the lysosomes compared to control cells both under basal conditions and following stress (Figure 5C). Specifically, in untreated PLEKHM2[delAG] cultures, lysosomes clustered together, while in control cultures the lysosomes were distributed farther from the nuclei. Post-treatment control neurons presented even greater dispersion of lysosomes, whereas lysosomal clusters in PLEKHM2[delAG] neurons remained mostly perinuclear. Overall, these results suggest that PLEKHM2 is involved in anterograde transport of lysosomes and that, in PLEKHM2[delAG] neurons, this transport is attenuated. These differences between the lysosomes in control and in PLEKHM2[delAG] cells may shed light on the observed differences in the electrophysiology and autophagic activities and agree with studies demonstrating that lysosome size, numbers and distribution impact neuronal functions [36,37,38,39].

## 3. Discussion

Autophagy dysregulation has been associated with the stimulation of neurodegenerative disease progression [14,17,24,28,40]. PLEKHM2 was linked to autophagy regulation in the context of cardiovascular disease [5], however its role in the context of neurodegeneration remains largely unknown. Clinical neurodegeneration has not been reported in homozygous PLEKHM2[delAG] patients, though it might be as a result of the premature death of these patients due to heart failure. Here, we used our previously established set of PLEKHM2[delAG] patient-specific iPSCs and their associated healthy heterozygous control, to interrogate the role of PLEKHM2 in neuronal development, maturation and function.

We first examined attributes of the mutation in PLEKHM2 on the development of iPSC-derived neural cultures and demonstrated that the PLEKHM2 mutation does not affect the differentiation potential of iPSCs into a neural lineage. These results suggest that PLEKHM2 is not involved in neural differentiation, in line with the lack of early neurological phenotypes in PLEKHM2[delAG] patients [5]. Cellular ability to regulate autophagosomes directly influences autophagic processes. Either accumulation or insufficient formation of autophagosomes is witnessed in various pathologies such as PD [41,42], AD [43,44], chronic kidney disease [45], ovarian cancer and melanoma [46]. Primary PLEKHM2[delAG] fibroblasts were previously shown to present reduced autophagic flux following leupeptin treatment [5]. Similarly, we demonstrated that PLEKHM2[delAG] NEP cultures also exhibit reduced autophagic flux. This could be due to mechanisms such as the inability to produce sufficient autophagosomes [47,48], failure of autophagosomes and autolysosomes to decompose their content at the required rate [49,50], or high levels of basal autophagosomes that interfere with proper autophagic induction—only generating a miniscule flux [51,52,53]; all these possibilities may affect cellular homeostasis. Such a deviation might be disastrous for neurons and has been proven to provoke neurodegeneration in AD, PD, ALS and FTD [17,20,41,43]. DCM-LVNC patients died before any neurological disorder was detected; it is known that many neurodegenerative diseases are age related [15,54] and therefore reduction in autophagic flux may have a cumulative effect, especially since *PLEKHM2* transcripts are highly expressed in the brain [5]. Activation of autophagy, on the other hand, has improved health and prolonged the life span of *Drosophila* [55], yeast [56] and even mice [57,58,59].

To test the possible role of PLEKHM2 in neuronal functions, we next interrogated the electrophysiological activity throughout the differentiation of iPSCs into iMNs on MEA plates. Electrophysiological analysis showed that both control and PLEKHM2[delAG] cultures contained active viable neurons, but they differed in their functional performance. PLEKHM2[delAG] iMN cultures displayed a delayed peak in spontaneous electrophysiological activity. Similar electrophysiological phenotypes have been previously linked to delayed neuronal maturation [60], thus, our results suggest a possible role for PLEKHM2 in functional neuronal maturation. Furthermore, PLEKHM2[delAG] neural cultures were highly active, their firing of action potentials were unsynchronized and showed many rapid consecutive spikes followed by quiescent short intervals between firing of additional action potentials. Additionally, action potentials in patient cell cultures rose considerably later than in control cultures. The dysregulation in PLEKHM2[delAG] cultures activity is reminiscent of the neural behavior in epilepsy [61,62,63], autism spectrum disorder (ASD) [64] or ALS/FTD [20]. Interestingly, despite elevated spontaneous activity in PLEKHM2[delAG] cells, the neuronal PLEKHM2[delAG] cultures displayed decreased synchrony within the neuronal network. These results suggest that PLEKHM2 plays a role in neuronal connectivity. Intercellular neuronal communication is mediated chiefly through chemical synapses [65,66], thus suggesting that PLEKHM2 may promote synaptic functions. Similarly, impaired neuronal communication has been linked to synaptic dysfunctions [67,68,69].

To find a link between the mutation in PLEKHM2, which functions as a mediator for lysosomal anterograde motion, and the impaired electrophysiological functions of the iPSC-derived neurons cells, we examined the lysosome size and distribution. PLEKHM2[delAG] neurons had increased lysosome size under both basal and stress conditions compared to the control neurons (Figure 6B(ii)). Enlarged lysosomes might indicate defective autophagic lysosomal reformation (ALR) [70]. Loss-of-function of alfa-1 (c9orf72 ortholog in *Caenorhabditis elegans*), which is related to lysosome regulation, has been found to damage ALR [20] and an increased expression of proteins participating in autolysosomal fusion, such as Rab7 [70,71], also discouraged reformation and resulted in larger lysosomes [70]. Interference in anterograde motion, which assists ALR by initiating tubulation in the separation of the autophagosome and the lysosome [70] may, as a result of mutated *PLEKHM2*, consequently generate enlarged organelles as shown in PLEKHM2[delAG] neuronal cultures. Larger lysosomes could be indicative of their acidity as shown, for example, in the case where an addition of weak basic compounds to a macrophage culture inflated the lysosomes [72]. The process involves alkalizing lysosomal pH by promoting absorption of water into the organelles and subsequently inhibiting lysosomal hydrolases activity. Since PLEKHM2[delAG] organelles are enlarged, it is plausible that they are less acidic and therefore less active in protein degradation [73,74]. Basal autophagy is crucial, especially in postmitotic cells such as neurons and cardiomyocytes [24,75,76] in which the irregularity stemming from defective ALR or less acidic lysosomes is potentially harmful to the cells. Lysosomal positioning also relates to the cellular state. Under normal conditions anterograde motion is upregulated, thus organelles should disperse through the cell. CQ simulates such conditions as it alkalizes lysosomal pH. However, under starvation, retrograde motion is upregulated and perinuclear displacement of autophagosomes and lysosomes is expected [73]. PLEKHM2[delAG] iMN cultures presented reduced distribution of lysosomes throughout the cells under basal conditions. R + CQ treatment dispersed the lysosomes in control cultures, while in PLEKHM2[delAG] cultures lysosomes remained perinuclear, suggesting that PLEKHM2[delAG] cells are experiencing stress even before R + CQ treatment (Figure 6). Lysosome positioning also controls autophagosome formation rate through the mTOR pathway; specifically, perinuclear placement encourages autophagosome genesis to favor auto-lysosomal fusion [77]. Indeed, both abnormal lysosomes size and localization observed in PLEKHM2[delAG] neurons suggest a mechanistic explanation for the impaired function of PLEKHM2[delAG] cultures.

Additionally, motor protein–vesicle complex formation proteins control much of the cellular distribution of vesicles. Interference with the BORC-ARL8-PLEKHM2-KINESIN-1 ensemble has been shown to downregulate the anterograde axonal transport of lysosomes, which is essential for axonal growth cone dynamics [21]. Lysosomes participate in synaptic maturation and neurotransmission, however, these neuronal processes do not rely solely on lysosomal function, but are supported by other vesicles such as pre-synaptic vesicle precursors (SVPs) carrying neurotransmitters and presynaptic active zone (AZ) proteins. A mutation in BORC resulted in lysosomal perinuclear positioning, while SVPs were still able to reach pre-synaptic terminals [9]. On the other hand, ARL8 decrease or loss-of-function halted both synaptic vesicle (SV) and AZ proteins’ anterograde movement [78,79,80]. Furthermore, such SVs were larger and mostly static in *C. elegans* motoneurons [9]. Consequently, all proteins involved in vesicle motility may affect interactions differently, but each performs a specialized role, which is particularly important in polarized cells such as neurons.

Furthermore, during neuronal and synapse maturation, lysosomes localize to the axons and dendrites, assisting with membrane remodeling through elongation and exocytosis [81,82,83,84]. The perinuclear localization in neurons of PLEKHM2[delAG] iMN cultures implies that synapse maturation did not initiate in these cells and therefore supports the latency in neural activity compared to the control lines. Moreover, larger lysosomes have been associated with less-active neural cultures, as they performed less exocytosis [85]. Downregulation of exocytosis can interfere with neural connectivity as neurotransmitters are secreted at a lower rate. For example, in Mucopolysaccharidosis type VII (MPS VII), an autosomal recessive lysosomal storage disease, neural cultures contained less astrocytes, which are known to regulate neurotransmitter metabolism, and presented low network connectivity [86].

Indeed, our study shows that while neurons in control iMN cultures were able to highly synchronize, PLEKHM2[delAG] neurons failed to do so. Furthermore, frequent spikes and high median ISI within bursts and burst percentage also suggest inconsistent exocytosis and neurotransmitter exertion rates due to faulty anterograde lysosomal movement. In *Atg7* deficient dopaminergic neurons, the reduction in autophagy leads to quicker presynaptic recovery and evoked dopamine release [87], resembling the results exhibited in this study for PLEKHM2[delAG] cells.

Collectively, our results suggest a model (Figure 6) in which lysosomes in PLEKHM2[delAG] cells do not scatter to the periphery, leading to the impairment of autophagy, downregulation of exocytosis and, consequently, a lower rate of neurotransmitter secretion. Altogether, this leads to abnormal neural connectivity. We therefore conclude that PLEKHM2-mediated anterograde lysosomal movement during neural development plays an important role in proper autophagy and, consequently, in functional neuronal maturation.

## 4. Materials and Methods

### 4.1. Cell Culture

iPSC lines were previously described in detail [23]. Cells were cultured in NutriStem hESC XF medium (Biological Industries, Kibbutz Beit Haemek, Israel) in feeder-free conditions on Matrigel-coated plates (BD Biosciences, San Jose, CA, USA). The pluripotency of the various iPSC lines was routinely assessed by flow cytometry analysis of the pluripotent marker OCT3/4 and a mycoplasma test (Hy-Mycoplasma PCR, hylabs) was performed monthly. All cells were cultured under standard conditions at 37 °C and 5% CO_2_, in a humidified incubator.

### 4.2. Neuronal Differentiation

Differentiation of iPSCs into motor neurons was performed as previously described with minor modifications [27]. iPSCs were cultured in NutriStem medium on Matrigel and passaged weekly at 80% confluence at a ratio of 1:6 to 1:15 using Versene (GIBCO, Gaithersburg, MD, USA). For differentiation generating neuroepithelium (NEP) stock, split iPSCs were seeded on 6-well Matrigel-coated plates at a density of 10^4^ or 1.5 × 10^4^ cells/cm^2^ for control and PLEKHM2[delAG] lines, respectively, since the latter required another day to reach confluence. Neural induction to generate NEP was initiated at 30–40% confluence: Stage 1 medium (S1M), containing Iscove’s modified Dulbecco’s medium (IMDM): F12 1:1, 2% B27, 1% N2, MEM non-essential amino acids solution (NEAA) and Penicillin–Streptomycin–Amphotericin B (PSA), 10 μM SB-431542 (SB), 3 μM CHIR-99021 (CHIR) and 0.2 μM LDN-193189 (LDN). Medium was replaced daily for 6 days. On day 6 (D6), cells were cryopreserved, following 5 min dissociation with Accutase (GIBCO, Gaithersburg, MD, USA). Cell pellets were generated at 250 g for 7 min and kept in freezing medium containing 90% fetal bovine serum (FBS) and 10% Dimethyl sulfoxide (DMSO). Stage 2 medium (S2M) contained S1M constituents with an addition of 0.1 µM all-trans retinoic acid (RA) and 1 µM smoothened agonist (SAG). On D6, single NEP cells were dissociated using Accutase for 5 min and seeded at 1.25 × 10^5^ cells/cm^2^ in S2M on 5 μg/cm^2^ laminin-coated wells. When thawed, cells were seeded at a concentration of 1.5 × 10^5^ cells/cm^2^ and 5 µM ROCK inhibitor (Y-27632) was added. For imaging purposes, cells were seeded at 5 × 10^4^ cells/cm^2^. Half media was exchanged every other day for 6 days. On D12, all medium was replaced with Stage 3 medium (S3M) to promote maturation. S3M medium contained IMDM: F12 1:1, 2% B27, 1% N2, NEAA and PSA, 2.5 µM γ-secretase inhibitor (DAPT), 0.5 µM RA, 0.1 µM compound E, dibutyryl cyclic AMP (db-cAMP) and SAG, 200 ng/mL ascorbic acid (AA) and 10 ng/mL brain-derived neurotrophic factor (BDNF) and glial-derived neurotrophic factor (GDNF). Half of S3M was exchanged every other day for the remainder of the differentiation.

### 4.3. Flow Cytometry Quantification

Cells were dissociated into a single-cell suspension using Versene for undifferentiated iPSCs or Accutase for differentiated cells and fixed using the Foxp3/Transcription Factor Staining Buffer Set (eBioscience, San Diego, CA, USA), according to the manufacturer’s instructions. Cellular marker quantification was performed using NovoCyte NovoSampler Pro (Acea Biosciences, Santa Clara, CA, USA), utilizing FlowJo^TM^ software (BD Life Sciences, Franklin Lakes, NJ, USA). Primary antibodies used were anti OCT3/4 (sc-2004, 1:100, Santa Cruz, Dallas, TX, USA), SOX1 (ab87775, 1:500, Abcam, Cambridge, UK), NESTIN (ABD69, 1:500, Sigma-Aldrich, Saint Louis, MO, USA), TUJ1α (T8660, 1:1000, Sigma-Aldrich, Saint Louis, MO, USA), GFAP (ab7260, 1:1000, Abcam) ISL1 (ab178400, 1:100, Abcam, Cambridge, UK), MAP2ab (M1406,1:500, Sigma-Aldrich, Saint Louis, MO, USA) or ChAT (GTX113164, 1:500, GeneTex, Irvine, CA, USA). Secondary antibodies were goat anti-mouse Alexa Fluor 488 (A-11029, 1:500, Thermo Fisher, Waltham, MA, USA), goat anti-Rabbit Alexa Fluor 633 (A-21071, 1:500, Thermo Fisher, Waltham, MA, USA).

### 4.4. Autophagy Flux Detection

For autophagy measurements, iPSCs and differentiated neural cultures were assessed using the CYTO-ID Autophagy detection kit (Enzo Life Sciences, Farmingdale, NY, USA), to identify autophagosomes in living cells according to the manufacturer’s protocol, similarly to [88]. Briefly, iPSCs were seeded in 12-well plates at a density of 4 × 10^4^ or 4.5 × 10^4^ cells/ cm^2^ for control and PLEKHM2[delAG], respectively. Two days after seeding, stress induction was applied. Differentiated cells were seeded according to the differentiation protocol described above and treated 1 day prior to the experiment. Stress treatments inducing autophagy consisted of 500 nM rapamycin and 20 μM chloroquine for 20 h or F12 (HAM) nutrient media (termed here nutrient limited media (NLM)) and 20 μM chloroquine for 4 h, DMSO was used as a vehicle control. The iPSCs and differentiated cells were detached using Versene and Accutase, respectively, washed with assay buffer X1 supplemented with 5% FBS and pelleted at 250 g for 5 min. The pellet was then incubated for 30 min at 37 °C, 5% CO_2_ with CYTO-ID Green Detection Reagent (1:2000), washed with assay bufferX1 supplemented with 5% FBS and analyzed using flow cytometry. Median fluorescence intensity (MFI) was calculated for each representative histogram, and the fold change was calculated by dividing results of treated by vehicle controls.

### 4.5. Microelectrode Array (MEA) Assay

At D6 NEP were plated on 24-well MEA plates (Axion BioSystems, Atlanta, GA, USA) coated with 0.1% polyethyleneimine (PEI) and 5 μg/cm^2^ laminin at a density of 1.25 × 10^5^ cells/cm^2^ or 1.5 × 10^5^ cells/cm^2^ when thawed and differentiated according to the protocol stated above (Section 4.2). Spontaneous activity was measured every 2–3 days for 15 min using a Maestro Edge apparatus (Axion BioSystems, Atlanta, GA, USA). Waveform events were identified using adaptive spike threshold crossing with a standard deviation (SD) of electrode noise set at 6. A minimum of five spikes per minute was used to include events for analysis.

### 4.6. Western Blot

Cells were collected for protein extraction in ice-cold radioimmunoprecipitation (RIPA) buffer (Cell Signaling, Danvers, MA, USA) supplemented with 1 mM phenylmethylsulfonyl fluoride (PMSF). Total protein quantification was performed using micro-BCA protein assay kit (Pierce Biotechnology, Waltham, MA, USA). Next, 20 μg total protein samples were separated on a mini-PROTEAN TGX gel (Bio-Rad, Hercules, CA, USA) for 75 min under 100 V current, and then transferred to a nitrocellulose membrane (Bio-Rad, Hercules, CA, USA) 45 min, 100 V. The membrane was incubated for 1 h in 5% bovine serum albumin (BSA) (Millipore, Bedford, MA, USA) blocking solution, followed by an overnight incubation with primary antibodies: anti-LC3B (nb100-2220, 1:500, Novus Biologicals, Littleton, CO, USA), anti-p62 (P0067, 1:500, Sigma-Aldrich, Saint Louis, MO, USA) and anti-GAPDH (G9545, 1:1000, Sigma-Aldrich, Saint Louis, MO, USA). Later, the membrane was incubated with a secondary antibody goat anti-rabbit IgG-HRP (sc-2004, 1:5000, Santa Cruz) for 1 h and then with EZ-ECL kit (20–500, Biological Industries, Kibbutz Beit Haemek, Israel). The signal was detected using Fusion Solo X (Vilber Lourmat, Collegien, France) and densitometric analysis was carried out using ImageJ software (U. S. National Institutes of Health, Bethesda, MD, USA, http://imagej.nih.gov/ij/, accessed on 3 May 2021). Band intensity was normalized to GAPDH.

### 4.7. Immunocytochemistry

Cells were fixed with 4% formaldehyde for 15 min at room temperature, washed and blocked for 1 h in 5% BSA (Millipore, Bedford, MA, USA) with 0.1% Triton-X 100 in PBS. Then, cells were incubated overnight with primary antibodies against LAMP1 (D2D11, 1:500, Cell Signaling Technology, Danvers, MA, USA) and TUJ1α (T8660,1:1000, Sigma-Aldrich Aldrich, Saint Louis, MO, USA) at 4 °C. Followed by a 2 h incubation at room temperature with donkey anti-rabbit Alexa Fluor 594 antibody (711-585-152, 1:500, Jackson ImmunoResearch, West Grove, PA, USA) and donkey anti-mouse Alexa Fluor 488 antibody (715-545-150, 1:500, Jackson ImmunoResearch, West Grove, PA, USA). After a brief wash, samples were stained with NucBlue (R37605, Themo Fisher Scientific, Waltham, MA, USA) for nuclei detection. Fluorescent images were acquired using Nikon C1si laser scanning confocal microscope abb. LSCM (Nikon, Tokyo, Japan) and EVOS FL (AMG, Bothell, WA, USA).

### 4.8. Biodistribution Assay

Coverslips were first coated with poly-L-ornithine 1 mg/mL overnight in an incubator, then with a drop of laminin 50 µg/mL for 1 hr. NEP cells were plated at a concentration of 2 × 10^6^ cells/mL and differentiated to an immature motor neurons (iMN) culture. Stress was induced with 500 nM rapamycin and 20 μM chloroquine for 20 h and the culture was stained for neurons (TUJ1α) and lysosomes (LAMP1). Image acquisition and analysis were performed similarly to a previous study [80] with minor modifications. Lysosomes were automatically identified according to size using the particle analysis tool in ImageJ. Those particles were manually attributed to neurons using the neuron image channel. Only fields in which neurons could be visually separated were taken. A minimum of seven fields were taken using a confocal microscope ×63 AyrScan Laser Scanning Microscopy with Super-Resolution (Zeiss LSM880 Airyscan) and analyzed using ImageJ. Particles ranging in a size of 0.008–1.2 μm^2^ were considered as lysosomes.

### 4.9. Statistical Analysis

Statistical analysis was performed with GraphPad Prism version 6.07 for Windows (GraphPad Software, San Diego, CA, USA). Results are presented as mean ± SEM of two clones per genotype (control or PLEKHM2[delAG]) from at least two independent experiments. Two-tailed student’s *t*-test and for repeated measures, distribution normality was tested using Kolmogrov–Smirnov test with the Dallal–Wiloksinson–Lilliefor corrected p value. Normal distributions were tested using one-way ANOVA with Sidak’s multiple comparison post hoc test. Data sets which were not normally distributed were analyzed using Kruskal–Wallis non-parametric test with Dunn’s multiple comparison post hoc test. Values of *p* < 0.05 were considered significant.

## Figures and Tables

**Figure 1 ijms-23-16092-f001:**
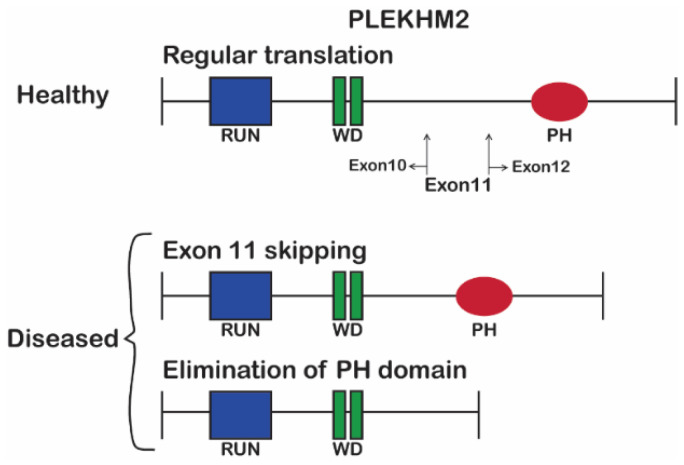
PLEKHM2 protein structure in healthy individuals (top panel) and the two possible variants that are truncated as a result of AG nucleotide deletion in DCM-LVNC patients (bottom panel).

**Figure 2 ijms-23-16092-f002:**
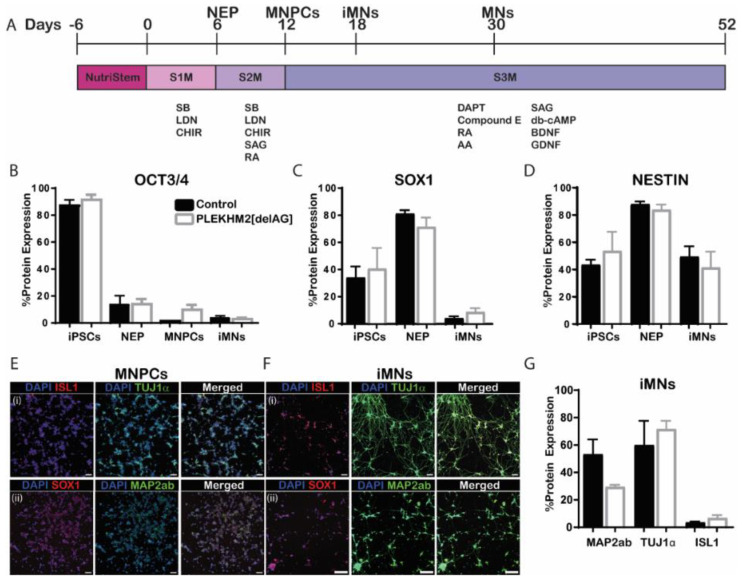
PLEKHM2 is not involved in neural differentiation potential. (**A**) Differentiation protocol layout, D6 neuroepithelium (NEP), D12 motor neuron progenitor cells (MNPCs), D18 immature motor neurons (iMNs) and from D30 on motor neurons (MNs). (**B**–**D**) Protein expression by flow cytometry analysis. Organized by row accordingly: the pluripotency marker OCT3/4 (**B**), the neural progenitor cell markers SOX1 (**C**) and NESTIN (**D**). One-way ANOVA with Sidak’s multiple-comparisons test was used. Data presented as mean ± SEM *n* = 2–6. (**E**,**F**) ICC of MNPCs (**E**) and iMNs (**F**) for ISL1, TUJ1α (**i**) SOX1 and MAP2ab (**ii**). Scale bars—100 µm. Blue—DAPI, red—ISL1, SOX1 and green—TUJ1α, MAP2ab. (**G**) Specific neuronal markers in iMN cultures—MAP2ab, TUJ1α and ISL1. One-way ANOVA with Sidak’s multiple-comparisons test was used. Data presented as mean ± SEM *n* = 2–6.

**Figure 3 ijms-23-16092-f003:**
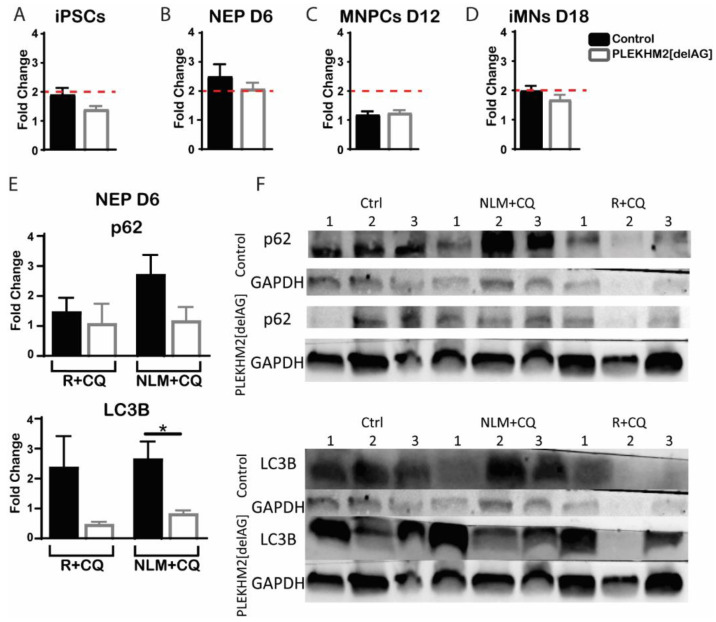
Reduced autophagic activity trend in PLEKHM2[delAG] cultures. (**A**–**D**), Autophagy flux acquired by flow cytometry analysis during four stages of differentiation using the CYTO-ID autophagy detection kit. iPSCs (**A**), NEP (**B**), MNPCs (**C**), and iMNs (**D**). Stress induction by rapamycin and chloroquine (R + CQ) treatment, unless stated otherwise. Black and empty bars represent control and PLEKHM2[delAG] cultures, respectively. Fold change calculated by median fluorescent intensity after treatment normalized to untreated samples. The dashed red line marks the 2-fold change. Two-tailed student’s *t*-test was employed, presenting mean ± SEM *n* > 3. (**E**) Western blot analysis of D6 culture autophagosome markers p62 and LC3B under two stress inductions, R + CQ and nutrient limited media with chloroquine (NLM + CQ). * *p* = 0.0155 by two-tailed student’s *t*-test. Bar graphs present mean ± SEM *n* = 5 (including 2–3 technical repetitions in at least two independent experiments for each line) (**F**) Representative Western blots. Autophagosomal markers, p62 and LC3B and housekeeping gene, GAPDH.

**Figure 4 ijms-23-16092-f004:**
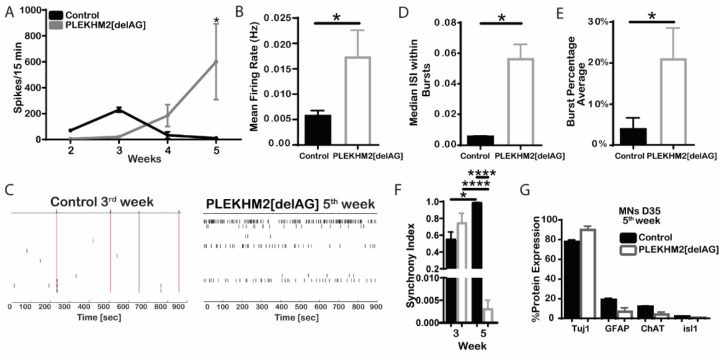
Microelectrode array (MEA) functional assay of the cultures. (**A**) Number of spikes fired during a 15 min recording. Based on at least four separate independent experiments (BGUi008A—2, BGUi008B—2, BGUi009—2 and BGUi010—3) * *p* = 0.025 by one-way ANOVA with Sidak’s multiple-comparisons test. (**B**) Mean firing rate * *p* = 0.0171 by two-tailed student’s t-test. Presenting mean ± SEM. (**C**) Representative raster plots during the most active week in the culture. (**D**) Median inter-spike interval (ISI) within bursts * *p* = 0.0214 by two-tailed student’s *t*-test. (**E**) Burst percentage average * *p* = 0.0345 by two-tailed student’s *t*-test. All graphs presenting mean ± SEM. (**F**) Synchrony index during weeks 3 and 5 * *p* = 0.0130, **** *p* < 0.0001, respectively, by two-tailed student’s t-test, showing mean ± SEM. Week 3: n_control_ = 7, n_PLEKHM2[delAG]_ = 3. Week 5: n_control_ = 2, n_PLEKHM2[delAG_] = 7. (**G**) Protein expression of day 35 neural culture. Two-tailed student’s t-test, showing mean ± SEM, *n* = 2.

**Figure 5 ijms-23-16092-f005:**
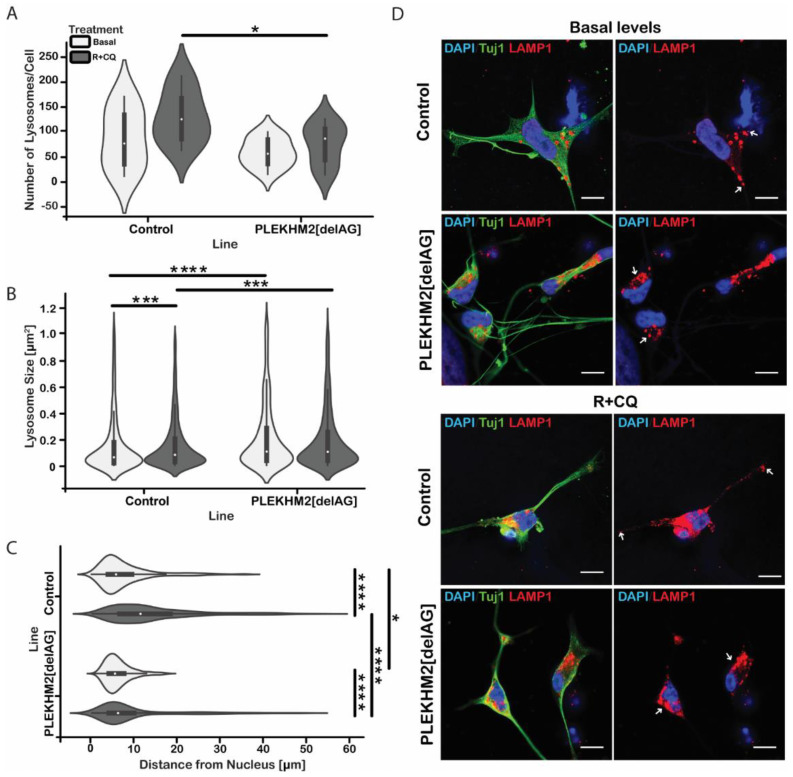
Lysosomal distribution. (**A**) Number of lysosomes within each cell. The interquartile range (IQR) is denoted by the black rectangle, white dots denote the median value * *p* = 0.0206 by one-way ANOVA Sidak’s multiple-comparisons test. (**B**) Lysosome size distribution in baseline and following stress induction. *** *p* < 0.007, **** *p* < 0.0001 by one-way ANOVA with Kruskal–Wallis multiple-comparisons test. (**C**) The distance of lysosomes from the nucleus is measured. * *p* = 0.0213, **** *p* < 0.0001 by one-way ANOVA with Kruskal–Wallis multiple-comparisons test. (**D**) Representative images of cells during basal conditions and following R + CQ treatment. White arrows point to lysosomes far from the nucleus in control cultures or perinuclear in PLEKHM2[delAG] cultures. *n* = 11–14 cells from two independent experiments. Scale bar—10 µm. Blue—DAPI, green—TUJ1α, red—LAMP1.

**Figure 6 ijms-23-16092-f006:**
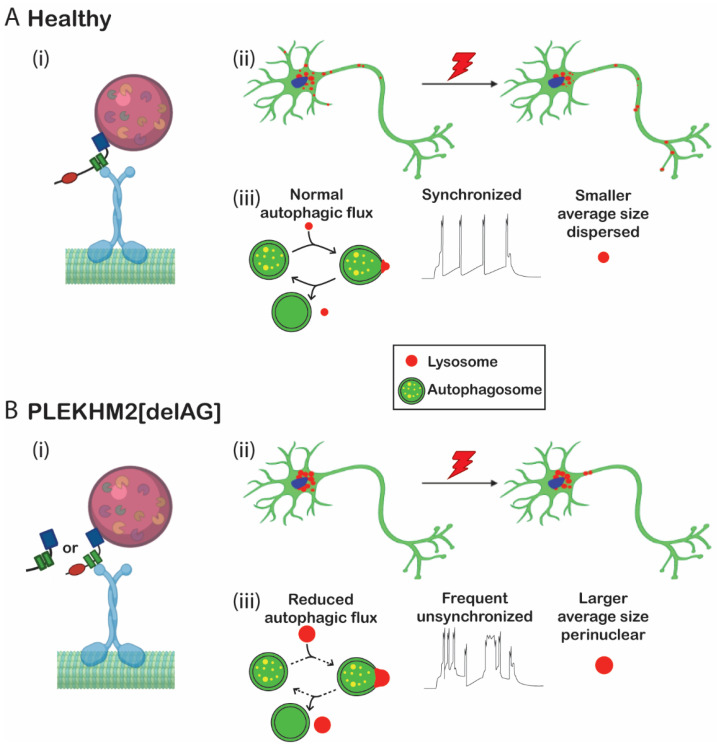
Suggested mechanism: large perinuclear lysosomes lead to abnormal electrophysiological activity in (**A**) healthy and (**B**) PLEKHM2[delAG] cells. (**i**) Lysosome binds to KINESIN-1 motor protein through PLEKHM2. Healthy control translation of the protein allows for the desired regulation of anterograde motion, while PLEKHM2[delAG] transcripts are translated into two shorter variants. The first skips exon 11 but all three domains are translated. Translation of the latter halts at the mutation site (AG deletion). The shorter variants probably interfere with the required anterograde motion. (**ii**) Lysosomal biodistribution under basal conditions and after stress induction. Translation of PLEKHM2 protein results in lysosomes spreading throughout the cell, while most remain in the soma. Applied stress signals the lysosomes to move anterogradely to the neurites in control cultures, whereas PLEKHM2[delAG] protein variants allow for a transient interaction between KINESIN-1 and the lysosome, and consequentially, a more perinuclear localization persists. Bigger lysosomes indicate either autolysosomal conformation or inactive alkalinized lysosomes. Stress induction allows a modest spreading of the lysosomes, without enlargement of the organelles. (**iii**) A summary of the differences recorded between the cultures. Control cultures presented a smaller average size of lysosomes which were synchronized and, after stress induction, had stimulated autophagic flux. m*PLEHM2* cultures had larger lysosomes which clustered by the nucleus under basal conditions and stress induction. These cultures were unsynchronized and fired frequently. The autophagic flux registered was lower than control cultures. Figure partially created with BioRender (https://biorender.com/, accessed on 13 December 2022).

## Data Availability

Data is contained within the article. Further details are available on request from the corresponding author.

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
