# Peer review of "PLEKHM2 Loss of Function Impairs the Activity of iPSC-Derived Neurons via Regulation of Autophagic Flux"

_ijms, 2022, doi:10.3390/ijms232416092_

Round 1

Reviewer 1 Report

This study addresses the role of the human PLEKHM2 gene, involved in cardiomyopathy, in motor neuron cultures derived from patient-specific and control iPSCs. They find that PLEKHM2 mutant cells displayed normal motor neuron differentiation in vitro but showed signs of reduced of reduced autophagic activity and unsynchronized electrical activity. While human disease modelling using iPSC platforms represent an important avenue of research, there are several shortcomings in the current study that raises concerns regarding its validity at the current stage.

Major issues:

1)      The logic behind the study is unclear. If patients homozygous for PLEKHM2[delAG] show no neurological phenotypes before they die as young adults, why would the authors believe that there is anything majorly wrong with their motor neurons? And that potential minor issues in motor neurons could be reliably uncovered in iPSC->iMN cultures?

2)      Figure 1, Main text, row 71: Ref.5 indicates that the shorter protein fragment of mPLEKHM2 was not detected in mutants, but rather a decreased level of total mPLEKHM2 was detected. Can the authors confirm if, in their cultures, mPLEKHM2 protein is either truncated and/or if protein levels are lower?

3)      Figure 2. Main text, row 83: “Motor neuron progenitor cells (MNPCs) continued to proliferate and extend neurites“. This is a confusing claim, since neural progenitors only begin to send out neurites after cell cycle exit. Further, there is no demonstration of cell proliferation or neurite extension in the cells, only FACS quantification of immunolabelling of proteins associated with these functions. The authors should either functionally demonstrate their claims for cell proliferation or neurite extension, or amend their text.

4)      Fig 2E: Map2a is a post mitotic neuronal marker, and it is therefore surprising to see it decrease as MNPCs differentiate into iMNs. Could the authors confirm 2E with immunohistochemistry of fixed cultured cells? If not, the validity of the staining is unclear.

5)      Fig 2F: Tuj1alpha is expressed in post-mitotic neurons, but not MNPCs. It is therefore surprising to see that there is a decrease in Tuj1a positive cells in iMNs compared to MNPCs. Could the authors confirm 2F with immunohistochemistry of fixed cultured cells? If not, the validity of the staining is unclear.

6)      Fig 2G: GFAP is an astrocytic marker and would not be expected to be expressed in either MNPCs or iMNs. Do the 2-4% of positive cells simply represent background staining? The authors could determine this by staining positive control cells from more mature neuronal tissue.

7)      Fig 2H: Isl1 is expressed in post-mitotic MNs, and not in their earlier progenitors. Thus, Fig 2H is confusing, since one would expect an increase in Isl1 positive cells in iMNs compared to NEPs. Could the authors confirm 2H with immunohistochemistry of fixed cultured cells? If not, the validity of the staining is unclear.

8)      Row 122. "These results are in line with the response observed in the flow cytometry analysis (Fig. 3A-B)". The authors previously stated that Fig 3A "showed insignificant differences" (row 104). Neither 3A nor 3B show significance. Which result are the authors arguing for? It is not clear that 3A and 3B show any significant differences.

9)      Fig 3. There is considerable variability between the Western blot triplicates e.g., in NLM+CQ and R+CQ for LC3B. What do the triplicates represent? The authors state that "iPSCs were seeded in 12 well plates", but it is not clear what the three replicates are i.e., are they from different wells of the same differentiation? Or different differentiations from different days? After clarifying this, can the authors increase their n to 5 or 6? This would bring confidence to the result.

10)   Fig 4. Can the authors clarify how many experiments were done to generate this figure? Was this a single differentiation, started on a single day for all three BGUi009, BGUi010 and BGUi008 lines? Or is it aggregated data from multiple differentiations started at different times? It is not clear that comparing between "the most active weeks" in culture is appropriate. Perhaps spiking synchrony of control cultures is equivalent to mPLEKHM2 cultures at 5 weeks, despite there being differences in spiking frequency. Can the authors please compare synchrony across all timepoints taken?

11)   What was the number of neurons in the control and mPLEKHM2 conditions for each week across the 5 weeks? Could the authors show a brightfield, low magnification wide-field image of both conditions for each week across the 5 weeks? The authors report percentage comparisons between control and mPLEKHM2, but not total cell yield. Total cell numbers could be different between the two conditions even though the ratios of the cell types are equivalent. This needs to be controlled by the authors, before concluding that "impaired functionality when compared to control cells." (row 134).

12)   Fig 5. Could the authors add an Isl1 staining in the far-red channel to confirm that these cells are motor neurons? Also, how was cell segmentation, and assignment of a lysosome to a nucleus, performed during the ImageJ analysis? While the results are interesting, it is not clear that the neurons are motor neurons, or how distant lysosomes were assigned to a nucleus. More detail on this needs to be added by the authors.

Minor issues:

13)   Nomenclature: The “mPLEKHM2” nomenclature is unorthodox and would result in mutations in all genes being referred to as “mSomething” when talking about mutant effects. The typical nomenclature is to refer to “cells/individuals mutated for PLEKHM2” or referring to the actual allele i.e., “cells/individuals carrying the PLEKHM2[delAG] allele show…”. Similarly, merging PLEKHM2 mutant iMNs into “mPLEKHM2-iMNs” is also an unusual nomenclature. I am not convinced that this manuscript is the place to change 100 years of genetic nomenclature standards.

14)   Graphical abstract: The "Differentiation" arrow should be removed from above the iPSCs and placed above the arrow leading to patient specific neural cultures.

15)   Text: Line 172: Should this read “truncated”, instead of “translated”?

16)   Row 107: "On the other hand, at the NEP stage, more than 2-fold increase in 107 autophagosomes was observed in response to treatment." Which figure is this referring to? 3B does not seem to have a significant 2-fold change.

Reviewer 2 Report

In this manuscript the authors investigate the neuronal role of PLEKHM2 using an iPSC model derived from healthy and DCM-LVNC patients .

1) Introduction

I think that the introduction should report a description of  the PLEKHM2/SKIP  role and function (For example:  Tal Keren-Kaplan and Juan S. Bonifacino, ARL8 Relieves SKIP Autoinhibition to Enable Coupling of Lysosomes to Kinesin-1, Current Biology November 23, 2020DOI:https://doi.org/10.1016/j.cub.2020.10.071). It is known that SKIP acts together with arl8 and kinesin-1 to transport lysosomes from the periphery to the center of the cell , we also know that arl8 and kinesin 1 regulate the anterograde transport  of axonal lysosomes (loss of arl8 induces the accumulation of lysosomes on the soma similarly to mutant skip).  Kinesin-1 requires SKIP to drive lysosome transport into axons (G.G. Farías et al., BORC/kinesin-1 ensemble drives polarized transport of lysosomes into the axon. Proc. Natl. Acad. Sci. USA, 114 (2017), pp. E2955-), and SKIP requires Arl8 in its active GTP-bound state (C. Rosa-Ferreira, S. Munro Arl8 and SKIP act together to link lysosomes to kinesin-1 Dev. Cell, 21 (2011), pp. 1171-1178)and finally overexpression of Arl8 in its active GTP-bound state (That can recruit skip) rescues the lysosomal accumulation on NPC -/- soma restoring lysosome transport into axons (Lipid-mediated motor-adaptor sequestration impairs axonal lysosome delivery leading to autophagic stress and dystrophy in Niemann-Pick type C Joseph C.. Developmental Cell Volume 56, Issue 10, 17 May 2021, Pages 1452-1468.e8).

2) Autophagy assays

The western blot  used to monitor the autophagy in control and mutant should evaluate the ration between LC3II and LC3I –as described in the  Guidelines for the use and interpretation of assays for monitoring autophagy (4th edition)1.

The authors claim thatPLEKHM2 Loss of Function  Impairs the Activity of 2 iPSC-derived Neurons via Regulation of Autophagic Flux”. To say that the author should demonstrate that skip regulate the autophagic flux looking the axonal maturation  and transport of autophagosomes and by measuring the lysosomes, autophagosomes fusion rate etc (confocal images etc). The authors shows only that loss of skip induces the accumulation of lysosomes and a defective autophagic flux (see number 3).

3) Lysosomes

The authors claim that PLEKHM2/skip  mutation lead to lysosome accumulation on the soma. Indeed, the structures were immunoreactive with anti-LAMP1 antibody, which led to the conclusion that PLEKHM2 mutations cause lysosome accumulation in the present and other reports in the literature. However, LAMP1 may also localize to other subcellular structures, such as endosomes. In fact a significant portion of LAMP1-labeled organelles do not contain detectable lysosomal hydrolases including cathepsins D and B and glucocerebrosidase.  Confirmation of these enlarged structures to be lysosomes with other independent lysosomal markers, such as LysoTrackers or lysosomal hydrolases, would strengthen the authors’ claim. (Xiu-Tang Cheng  et al.,  Characterization of LAMP1-labeled nondegradative lysosomal and endocytic compartments in neurons. J Cell Biol 2018 Sep 3;217(9):3127-3139. doi: 10.1083/jcb.201711083. Epub 2018 Apr 25).

4) Electrophysiological activity of iPSC should be discussed taking into consideration other work  Raffaella De Pace  et al,. Synaptic Vesicle Precursors and Lysosomes Are Transported by Different Mechanisms in the Axon of Mammalian Neurons. Cell Rep. 2020 Jun 16;31(11):107775. doi: 10.1016/j.celrep.2020.107775. (M.P. Klassen et al.,  An Arf-like small G protein, ARL-8, promotes the axonal transport of presynaptic cargoes by suppressing vesicle aggregation Neuron, 66 (2010), pp. 710-723;  Maeder et al., 2014; Anela Vukoja et al.,  Presynaptic Biogenesis Requires Axonal Transport of Lysosome-Related Vesicles Volume 99, Issue 6, 19 September 2018, Pages 1216-1232.e7. Neuron)

5) Please described better the “Autophagy Flux Detection”   in the method section  (see this manuscript as an example  Vantaggiato et al, Rescue of lysosomal function as therapeutic strategy for SPG15 hereditary spastic paraplegia. Brain. 2022 Aug 27:awac308. doi: 10.1093/brain/awac308.)

Round 2

Reviewer 1 Report

The logic behind studying this syndrome in motor neurons could still be articulated better.

Reviewer 2 Report

I'm satisfied of the revison. The manuscript is ready for publication.